# Changes in Estimating the Wild Boar Carcasses Sampling Effort: Applying the EFSA ASF Exit Strategy by Means of the *WBC-Counter* Tool

**DOI:** 10.3390/v14071424

**Published:** 2022-06-28

**Authors:** Stefano Cappai, Ileana Baldi, Pietro Desini, Antonio Pintore, Daniele Denurra, Marcella Cherchi, Sandro Rolesu, Daniela Mandas, Giulia Franzoni, Mariangela Stefania Fiori, Annalisa Oggiano, Francesco Feliziani, Vittorio Guberti, Federica Loi

**Affiliations:** 1Osservatorio Epidemiologico Veterinario Regionale (OEVR), Istituto Zooprofilattico Sperimentale della Sardegna, 09125 Cagliari, Italy; stefano.cappai@izs-sardegna.it (S.C.); sandro.rolesu@izs-sardegna.it (S.R.); daniela.mandas@izs-sardegna.it (D.M.); 2Department of Cardiac Thoracic Vascular Sciences and Public Health, University of Padova, 35131 Padova, Italy; ileana.baldi@unipd.it; 3ATS Sardegna, ASSL Sassari, Servizio di Sanità Animale, 07100 Sassari, Italy; pietro.desini@atssardegna.it (P.D.); danieledenur@virgilio.it (D.D.); 4Department of Wildlife, Istituto Zooprofilattico Sperimentale della Sardegna, 07100 Sassari, Italy; antonio.pintore@izs-sardegna.it (A.P.); marcella.cherchi@izs-sardegna.it (M.C.); 5Department of Animal Health, Istituto Zooprofilattico Sperimentale della Sardegna, 07100 Sassari, Italy; giulia.franzoni@izs-sardegna.it (G.F.); mariangela.fiori@izs-sardegna.it (M.S.F.); annalisa.oggiano@izs-sardegna.it (A.O.); 6Italian Reference Laboratory for Pestivirus and Asfivirus, Istituto Zooprofilattico Sperimentale dell’Umbria e delle Marche, 06126 Perugia, Italy; f.feliziani@izsum.it; 7Institute for Environmental Protection and Research (ISPRA), 00144 Roma, Italy; vittorio.guberti@isprambiente.it

**Keywords:** African swine fever, freedom of infection, passive surveillance, risk factor, wild boar, carcasses, exit strategy

## Abstract

African swine fever (ASF) is a devastating disease, resulting in the high mortality of domestic and wild pigs, spreading quickly around the world. Ensuring the prevention and early detection of the disease is even more crucial given the absence of licensed vaccines. As suggested by the European Commission, those countries which intend to provide evidence of freedom need to speed up passive surveillance of their wild boar populations. If this kind of surveillance is well-regulated in domestic pig farms, the country-specific activities to be put in place for wild populations need to be set based on wild boar density, hunting bags, the environment, and financial resources. Following the indications of the official EFSA opinion 2021, a practical interpretation of the strategy was implemented based on the failure probabilities of wrongly declaring the freedom of an area even if the disease is still present but undetected. This work aimed at providing a valid, applicative example of an exit strategy based on two different approaches: the first uses the wild boar density to estimate the number of carcasses need to complete the exit strategy, while the second estimates it from the number of wild boar hunted and tested. A practical free access tool, named *WBC-Counter*, was developed to automatically calculate the number of needed carcasses. The practical example was developed using the ASF data from Sardinia (Italian island). Sardinia is ASF endemic from 43 years, but the last ASFV detection dates back to 2019. The island is under consideration for ASF eradication declaration. The subsequent results provide a practical example for other countries in approaching the EFSA exit strategy in the best choices for its on-field application.

## 1. Introduction

Domestic and wild suids are the target susceptible populations of African swine fever (ASF), a highly contagious viral hemorrhagic disease caused by the ASF virus (ASFV) [1,2,3,4,5]. Considering its massive spread in recent years, threatening the global pig industry, ASF has earned a reputation for being potentially the most devastating disease of pigs [6].

The ASFV can be transmitted by direct or indirect contact (i.e., the dissemination of contaminated food or equipment); thus, infected wild boar constitute a major threat for domestic pig holdings [7]. Furthermore, the ASFV is highly stable under a wide range of environmental conditions [8], and indirect transmission through contact with infected carcasses is considered more likely than direct contact with live infectious animals [9]. Therefore, considering that the surveillance of wild species is much more difficult than that of domestic pigs [10], a combined approach of both active and passive surveillance of wild boar is of vital importance for ASF eradication, particularly when the virus prevalence and wild boar density are low when nearing eradication [11,12].

The recently published EFSA scientific opinion proposes a valid ASF exit strategy based on the passive surveillance of wild boar. This opinion aims to define standardize surveillance measures as a tool to provide robust evidence of the absence of ASFV circulation in wild boar [9]. On the basis of this opinion, each country in the last phases of ASF eradication should elaborate its specific strategy, considering the epidemiological context, choose the most appropriate time span for the screening phase, establish the necessary time of the confirmatory phase, and estimate the number of carcasses to be found in order to demonstrate the free status. Lastly, the overall process of the EFSA exit strategy strictly depends on the detection of subadult seropositive animals, which completely nullify the entire process, bringing the strategy back to the starting point [9].

On the other hand, implementing well-planned surveillance is not quite as simple, free from neither obstacles related to the amount of costs, the possible match of seropositive animals during the last phases of eradication, nor the need for a robust estimate of wild boar density to establish the number of carcasses to be detected. Examples of passive surveillance protocols are very rare. Desvaux et al. in 2021 [13] described a protocol applied on the border between France and Belgium aimed at assuring early detection in the case of the introduction of the disease and to support the free status of the level III risk area. Even if the implementation of the passive surveillance system contributed to the ASF-free status that Belgium regained in November 2020, it turned out to be highly expensive in term of time and human resources. The employment of 2769 manhours at a total cost of more than EUR 250,000 was necessary to completely cover the total level III risk area (9681 ha), leading to the detection of a total of six carcasses.

Moreover, even if the detection of seropositive animals during the last phases of ASF eradication is almost obvious [14], their epidemiological evaluation is extremely difficult. Several factors bias could affect their interpretation, such as the duration of the protective immunity in animals surviving from ASF and large variability associated with the duration of maternal antibodies [15,16,17,18,19,20]. Furthermore, the duration of the maternal antibodies developed in wild boar piglets may depend on an animal’s weight [21]. Thus, finding maternal antibodies may occur for a longer period than expected, increasing the variability associated with seropositivity findings [21].

Furthermore, in such contexts with low wild boar density, not enough animals are shot during hunting season to reach the necessary confidence level to find the virus at a very low prevalence (<1%) [22]. The implementation of passive surveillance that is effective enough to prove the absence of virus circulation strictly depends on a well-estimated animal density or a representative hunting bag. These measures are crucial to estimate the total number of expected wild boar carcasses in a specific area. Indeed, the passive surveillance feasibility depends on a country’s environment (i.e., vegetation, climate, and mountainous terrain). Good collaboration between hunters and veterinary services is mandatory to lead to greater efficacy in ASF detection [23].

This work aimed at providing a practical interpretation of the strategy based on the failure probabilities of wrongly declaring the freedom of an area even if the disease is still present but undetected. A valid applicative example of the exit strategy using Sardinian data on passive surveillance was described based on two different approaches: in the first approach, the number of carcasses required to complete the exit strategy is estimated by the wild boar density. The second approach starts from the number of wild boar hunted and tested during the hunting season. A practical free access tool, named *WBC-Counter* (available at: http://r-ubesp.dctv.unipd.it/shiny/WBC-counter/, last access 25 June 2022), was developed to automatically calculate the number of needed carcasses. The subsequent results will be of help not only to speed up the eradication of the disease in Sardinia, but also to provide a practical application for other countries approaching the EFSA exit strategy in the best choices for its on-field application.

## 2. Materials and Methods

### 2.1. Sardinian ASF Epidemiological Context

As reported by several previous studies [23,24,25], the ASFV has been endemic in Sardinia for more than 43 years. The Italian island has been affected by ASFV genotype I. In January 2022, ASFV genotype II was detected in wild boar carcasses recording about 200 wild boar cases in the north (Piedmont and Liguria regions) and in the middle (Lazio region) of Italy during the subsequent six months [26], and the first outbreak in a domestic pig farm in May 2022 [27].

It has been described that several socioeconomic factors, such as the breeds of the few pigs in small backyards and the presence of illegal free-ranging pigs, favored ASFV persistence on the island. Several control measures have been put in place since 2015: biosecurity in domestic pig farms was reviewed, including a double fence inside the infected area, controls to verify compliance with pig identification, intensifying registration, and carrying out massive culling actions of free-ranging pigs [28]. These measures drastically reduced the ASFV prevalence: the last virus detection in domestic pigs dates back to 2018, and to 2019 in wild boars. Nevertheless, ASFV seropositive (Ab+) animals are still detected. Recently, Sardinia’s status in relation to restrictions on ASF control measures changed, passing from the level of risk for countries included in Part IV (highest risk) to Part III of ASF risk areas (European Commission Implementing Regulation 2021/605/EU).

### 2.2. ASF Management in the Sardinian Wild Boar Population

Even if the island has been affected by the plague of ASF infection for several years, the disease has always been limited to an infected zone. The recently updated infected area surface (2021–2022 infected area) accounts for 5302 km^2^ and includes 62 municipalities. Wild boar hunting is managed differently inside and outside the infected zone, based on the limits of the wild boar hunting management units (HMUs). The HMU limits (Figure 1) are defined by the presence of natural or artificial barriers (i.e., rivers, mountains, and main roads), which prevent the movements of wild boar, hypothesizing separated metapopulations in each HMU [14]. 

Currently, active surveillance is mainly applied by hunters during the hunting season, regulated by article 18 of the National Italian Low n. 157 of 1992 [29], and most of the samples are concentrated during the winter season (i.e., November–January), as in most of the countries [9,12,30,31,32]. All of the wild boar hunted inside the infected area are tested for ASFV by a real-time polymerase chain reaction (RT-PCR), and serological tests are carried out by an Enzyme-Linked Immuno-Sorbent Assay (ELISA) as screening test, with confirmation via immunoblotting. In order to guarantee the adequate geographical distribution of samples based on the wild boar density and overall representativeness of the sampling outside the infected area, the authorities defined a sample size to be taken in each HMU to achieve the desired sensitivity of the surveillance (detection of at least one positive animal at a minimum prevalence of 5% with a 95% confidence level).

The passive surveillance activities were sped up since the end of 2019 via target actions put in place for searching for carcasses inside the infected area. Public awareness and warnings were disseminated through all of the involved stakeholders (i.e., veterinarians, Forest Corps, hunters, mushroom seekers, and climbers), with the aim of reporting all wild boar carcasses (entire or decomposed) to the authorities. All those found dead were subjected to a RT-PCR test, following the ASF official manual [33]. Data from these activities were collected in the Italian National Informative System (SINVSA). Furthermore, considering that the hardest Mediterranean vegetation in Sardinia is completely different to that of Northern European countries considered in the EFSA scientific opinion (i.e., Estonia) [9], a research project founded by the Italian Ministry of Health, aimed at defining the best carcass search protocol for the Sardinian environment, was implemented. The target areas with the higher chance of carcass detection were identified using a mathematical model [34]. Until the end of May 2022, a total of seven filed sessions were organized to investigate those target areas where seropositive wild boar were indeed recently found. Furthermore, since 2020, the competent authority decided to consider only found dead animals for passive surveillance and wild boar killed in road traffic accidents as active surveillance in combination with hunted animals to cover at least partially the nine months of missing data. This decision arose from the detection of similar expected prevalence between live animals killed during hunting and live animals killed by road traffic accidents [14].

### 2.3. The Key Points of the EFSA Scientific Opinion, Based on Estonian Data 

The proposed exit strategy, based on the mathematical model by Lange et al., 2021 [35], starts from the last ASFV detection and includes two phases, screening and confirmatory phases, complementary to each other.

In order to evaluate the country-specific context, the main key points of the EFSA scientific opinion were summarized [9]. To test if the context fulfilled the conditions of an ASF-free status declaration based on this opinion, eight indicators were developed (Table 1).

As specified in the scientific opinion, “the Exit Strategy recommendations were formulated per 1000 km^2^ but should be applied to the specific region size” [9]. A graphic tool was provided by the EFSA in order to evaluate the time span of the screening and confirmatory phases, based on the percentage of failure accepted, i.e., obtaining a false-negative result by proposing freedom from ASF while (undetected) infectious objects were still present in the area p. 52, Figure 24 [9].

Based on wild boar density data from Estonia, during the screening phase, the minimum number of carcasses to be found in order to increase the accuracy of the exit strategy and reduce the probability of a false exit decision is 1 per 1000 km^2^. The duration of this phase depends on the percentage of failure accepted (i.e., 2%, 5%, or 10%). During the confirmatory phase, enforced passive surveillance aiming to detect at least two carcasses/1000 km^2^/year must be applied. The greater the intensity of carcass collection chosen, the shorter the monitoring period of the confirmation phase has to last. Both the Exit strategy phases strictly depend by the level of confidence chosen (i.e., 2%, 5%, or 10%). If one seropositive subadult wild boar is detected, the strategy requires going back to the start of the screening phase. All samples collected by active surveillance must be tested for the ASFV, even if the secondary role and limited impact of active surveillance within the broader exit strategy were highlighted [9].

### 2.4. Standardization of the Exit Strategy and WBC-Counter Tool Development

Given the very low wild boar density in Estonia (0.3 wild boar/km^2^) and the even greater need to accurately apply the strategy, a standardized procedure was developed and implemented in the *WBC-Counter* tool. In order to standardize the number of carcasses to be found in a 1000 km^2^ area, two different approaches were developed, as described below.

#### 2.4.1. Wild Boar Density Approach

The wild boar density, expressed as number of animals/km^2^, was used to estimate the number of the wild boar population living in an area:$$wb_{pop}=wb_{dens}*1000\;{\mathrm{km}}^{2}$$

Assuming that 45% of the overall wild boar population is hunted [14,36,37,38], the amount of wild boar that die by being hunted is calculated as:$$wb_{hunt}=wb_{pop}*0.45$$

As suggested by the EFSA opinion, the *wb_hunt_* could be assumed as 90% of the overall dead wild boar, assuming that the natural mortality rate is 10%. Thus, the number of wild boar that naturally died (carcasses) was estimated as:$$wb_{nat\_d}=\left(\frac{\left(wb_{hunt}*100\right)}{90}\right)*0.1$$

The probability of finding carcasses during simple passive surveillance (the screening phase) was established by the EFSA opinion as 1% of the overall carcasses. Alternatively, during the confirmatory phase, a doubling of the effort is required, with the aim of finding at least 2% of the total carcasses.

#### 2.4.2. Wild Boar Hunted Approach

The number of wild boar hunted in 1000 km^2^ was assumed as the 90% of the overall dead wild boar, and the number of wild boar that naturally died (carcasses) was estimated as: $$wb_{nat\_d}=\left(\frac{\left(wb_{hunt}*100\right)}{90}\right)*0.1$$

As above, the probability of finding carcasses was established as 1% during the screening phase and at least 2% during the confirmatory phase.

In both of the approaches, the number of carcasses to be found was rounded up based on the surface area and time (months), as follows:$$\mathrm{Surface}\mathrm{proportion}:\;c_{exp}=\frac{Area*\hat{c}}{1000}$$
where *c_exp_* is the proportion of carcasses to be found within a specific *Area* (km^2^) based on the $\hat{c}$ estimated number of carcasses to be found in 1000 km^2^.
$$\mathrm{Time}\mathrm{proportion}:\;c_{exp}=\frac{Time*\hat{c}}{12}$$
where *c_exp_* is the proportion of carcasses to be found within a specific *Time* (months) based on the $\hat{c}$ estimated number of carcasses to be found during 1 year.

### 2.5. Sardinian Exit Strategy

As a practical example, the exit strategy was implemented using Sardinian data on passive surveillance of the 2021–2022 infected area, populated by an estimated number of 21,208 wild boar (wild boar density = 4/km^2^), and starting from the last virus detection (PCR+). In order to facilitate its application, ensuring the suitable distribution in time and space of the carcasses, the exit strategy was separately applied to each HMU included partially or completely in the wild boar-infected area in 2021–2022 (Figure 1). 

The specific Sardinian strategy was implemented based on the probability of a false exit decision and the time span chosen for both the screening and confirmatory phases, aiming to ensure the lowest probability of failure (2% probability of wrongly declaring the freedom of the island even if the disease is still present but undetected). In view of the uncertainty about natural mortality rates in Sardinia, a natural mortality of 10% and a mortality due to hunting of 90% were considered.

All of the spatial analyses were performed using ArcGIS^®^ 10.4.1 (ESRI, Redlands, CA, USA). The statistical analyses were carried out by the use of the open-source R-Software 4.1.0 (R Development Core Team, Vienna, Austria).

## 3. Results

The epidemiological data confirm the decline in the presence of ASF in the wild boar population already described [14,39]. Since 9 April 2019, when the last ASF cases were notified in two wild boar carcasses, the competent authority no longer detected the ASF virus genome in either wild boar or domestic or free-ranging pigs. Table 2 describes the ASF seroprevalence during the last three years from April 2019 to June 2022.

All wild boar cases detected during 2019–2021 were located inside the infected area mainly clustered in two areas (north and south), as graphically represented in Figure 2. The cases located in the first visual cluster in the north of the infected area were close to nature parks where hunting is not permitted (blue limits). The cases in the second visual cluster (south) were mainly detected in areas where the presence of illegal free-ranging pigs had been frequently detected.

As previously described, until 2020, the surveillance of wild boar shows the temporal clustering of the sampling, with most samples taken during the hunting season [14]. Statistical cluster analysis demonstrated that the sampling generally corresponded to the density of the wild boar population in the sampling areas [40].

As reported in Table 3, during the last three years (April 2019–June 2022), a total of 98 carcasses and 646 wild boar killed by road traffic were found inside the infected zone and tested negative for the ASF genome.

Figure 3a,b illustrates the geographical distributions of the samples taken with passive surveillance (yellow dots) and from road kills (color codes) in 2020 and 2021, respectively.

The activities of passive surveillance carried out between May and September (mean temperature of 16 °C) in those areas identified by the mathematical model [34] involved a total human involvement of 150 hours to investigate 40 hectares during passive surveillance activities on field. The areas were characterized by a Mediterranean cork forest with low visibility. Most parts of these areas were impenetrable to humans, and nonexaminable elements were present in the soil (i.e., thorns, briers, and rocky areas). Only two carcasses were found during these activities and several problems were faced given the rough vegetation, the availability of people, and the typical Sardinian climate. These preliminary results will be a valid starting point for future on-field activities programmed from February to June 2022 and 2023. All the other carcasses found during 2019–2021 arose from Forest Corps patrols, hunters, or citizens that voluntary called the Forest Corps. 

The three HMUs that include the wild boar infected area 2021–2022 are those historically named Goceano-Gallura (HMU-GG), Nuoro-Baronia (HMU-NB), and Gennargentu-Oglistra (HMU-GO). The part of the infected area surface included in each HMU is equal to 1716 km^2^ in Goceano-Gallura, 1189 km^2^ in Nuoro-Baronia, and 2497 km^2^ in Gennargentu-Oglistra (Figure 1). 

Based on the wildlife management plan and explained in previous studies [23,39,40,41], an estimated wild boar density equal to four animals/km^2^ was applied in *WBC-Counter* to estimate by the wild boar density approach the number of carcasses to be found in each HMU during the two exit strategy periods. The second wild boar hunted approach was assessed using the number of hunted wild boar divided by HMU. Data estimated by *WBC-Counter* are reported in Table 4 and Table 5. 

Based on a wild boar density equal to 3.5 animals/km^2^, an estimated population of 21,208 wild boar live in the overall 5302 km^2^ of the infected area. Of these, in order to complete the exit strategy, a total of 10 and 20 carcasses are needed during the screening and confirmatory phases, respectively. The carcasses must be overdispersed in time and space: during the screening phase, three must be found in GG-HMU, two in NB-HMU, and five in GO-HMU; a doubling of intensity is expected during the confirmatory phase, with 6, 4, and 10 carcasses, respectively. As illustrated in Figure 4, considering an overall population of between 1000 and 3000 carcasses alongside the total area, the collection of 30 ASF-negative carcasses corresponds to a sampling fraction equal to 1% and would be enough to exclude a prevalence lower than 9.5% in the carcasses with a confidence level of 95%.

Based on the number of total wild boar hunted during the hunting season (about 4500 wild boar inside the infected area), in order to complete the exit strategy, a total of five carcasses per year are needed during the screening phase: two ASF negative carcasses must be found in GG-HMU and GO-HMU, as opposed to one carcass in NB-HMU. From the doubling intensity effort, 10 ASF negative carcasses are needed during the confirmatory phase, subdivided into four carcasses in GG-HMU, two in NB-HMU, and four in GO-HMU (Table 5). As illustrated in Figure 4, the collection of 15 ASF-negative carcasses corresponds to a sampling fraction equal to 0.5% and would be enough to exclude a prevalence lower than 19% in the carcasses with a confidence level of 95%.

The complete data used for the exit strategy assessment by HMU are reported in Appendix A. As reported in Table 6, based on the wild boar density approach and considering the 2% probability of wrongly declaring the freedom of the island even if the disease is still present but undetected, the carcasses collected in GG-HMU during both the phases (9 months of screening and 10 months of confirmation) correspond to or are higher than those expected (two and six carcasses, respectively). In NB-HMU, the number of carcasses collected during the screening and confirmatory phases is equal to or higher than that expected (two and four carcasses, respectively), as well as in GO-HMU (three and eleven carcasses, respectively). Thus, the exit strategy can be declared as completed.

As reported in Table 5, in order to complete the exit strategy, a total of four and eight carcasses are needed during the screening and confirmatory phases, respectively. Considering an overall population of between 500 and 1000 carcasses alongside the total area, the collection of a sample of 12 ASF-negative carcasses corresponds to a sampling fraction equal to 2.4% and would be enough to exclude a prevalence lower than 22% in the carcasses with a confidence level of 95% (Figure 4). 

Table 7 illustrates the results arising from the wild boar hunted approach and considering the 2% probability of wrongly declaring the freedom of the island even if the disease is still present but undetected. The number of carcasses collected in GG-HMU, NB-HMU, and GO-HMU during the 9, 8, and 7 months of the screening phases, respectively, corresponds to or is even higher than that expected. In a similar fashion, the total number of carcasses found during the overall months of the confirmatory phase are enough to declare the exit strategy completed. 

## 4. Discussion

Uncertainty about the question of which kind of surveillance was more efficient in detecting the virus was still open until a few years ago. Recently, the key role of passive surveillance in the ASF eradication process was demonstrated [9,10,11], and several recommendations about its importance have arisen from national and international authorities [42]. This kind of surveillance is particularly recommended to detect the virus when a very low prevalence is hypothesized (i.e., the absence of virus detection) and in conditions of low wild boar density [11]. Furthermore, in the last phases of the disease eradication the surveillance activities should be designed to completely exclude the presence of residual pockets of the virus and demonstrate the absence of the circulation of the ASF virus [43].

On the other hand, the practical application of a passive surveillance program is not obvious nor easy to implement. The need for robust information concerning environmental conditions and animal density can lead the success of this surveillance [11], and a specific cost/benefit evaluation is mandatory. 

The recent EFSA exit strategy emphasized the importance of this kind of surveillance, in particular during the last phase of the ASF eradication process, and provided the first guidelines for its application [9]. On the other hand, the EFSA exit strategy was based on Estonian wild boar density (0.3 wild boar per km^2^), which is ten times lower than the wild boar density of most European countries [14,30,31,32]. Thus, each country that aims at implementing the exit strategy should adapt its application based on its own context. This is the first work aimed at standardize the approach providing practical guidelines and giving practical example on the application of ASF Exit strategy elaborated by EFSA. The online free tool *WBC-Counter* was developed in order to guide the choice of the best approach to all of these countries or region which are facing to the ASF eradication. Considering that some areas are characterized by more robust hunting data rather than wild boar density estimation and vice versa, two different background options are provided in the *WBC-Counter* tool based on the two different approaches are proposed in application to the EFSA Exit Strategy. The first based on the estimated wild boar density aims at collecting at twice the number of carcasses respect to the second approach based on the hunting bag. 

These different approaches bring attention to which kind of data should be used for the abundance of host species, given that these data strictly determine the results. The data on hunting bags are very sensitive to the efforts and management of the hunting season, while the use of data on environmental suitability and numerical indices are difficult to pass, especially in the medium–long period [44,45]. Sardinian passive surveillance data were used to develop a practical application of these two approaches. Based on the first approach, at least six more ASF-negative carcasses are needed to exclude a prevalence lower than 10% in the carcasses with a confidence level of 95%, completing the ASF Sardinian eradication. Otherwise, based on the second approach, the collection of 15 dead wild boar would provide less evidence for the absence of ASF in carcasses (a prevalence lower than 18% with a confidence level of 95%).

The results of the Sardinian surveillance program indicate considerable improvement in the epidemiological situation, as previously demonstrated [14,39,46]. Despite the clear decline in the seroprevalence of ASF, the sample intensity, mainly based on active surveillance limited to the hunting season, does not allow for the complete exclusion of the presence of residual pockets of the virus within the infected area, particularly in locations close to protected reservoirs, where hunting is not permitted [14]. Thus, the current robust data from passive surveillance are essential to finally exclude the disease in wild boar carcasses. In any case, the active surveillance data, mainly concentrated during the winter season, reflect the data from different European countries, where the bulk of the hunting bags is mainly collected between November and February [9,30,31,32]. Furthermore, the epidemiological situation around the nature parks where hunting is not permitted could not be ignored: these areas represent safe locations for wild boar during hunting activities [44]. Indeed, recently the importance of passive surveillance is even more clear to the veterinary authorities, which are working to increase the sensitivity of passive surveillance [9,11,14,42,47].

Despite the complete absence of the ASFV since 2019, the ongoing detection of seropositive wild boar put the island in a deadlock with the eradication of ASF, as established by the EFSA exit strategy. Updated Sardinian ASF data to 15th June 2022 (Table 2) highlight that during the finalization of the exit strategy, seropositive subadult wild boar were still detected. Thus, based on the exit strategy guidelines, the area should restart the process with the screening phase. Thus, the limits of serology should be further evaluated, and some gaps need to be addressed.

First, given that the detection of subadult seropositive animals completely nullifies the process, this is strictly related to the expertise of and collaboration with the hunters who have to date the wild boar. Furthermore, the EFSA opinion [9] defines subadults as those animals with age between 8 months and 2 years ± 6 weeks. Considering the Sardinian data, this definition partially includes the class subadult wild boar hunted (6–18 months). This could generate several biases, as well as the dentition itself, which is a proper feature of wild boar. Indeed, the cohort of wild boar births is longer than the hunting season, making dating to the fixed hunting period difficult and generating a considerable probability of error in addition to increasing the variability. Moreover, as underlined above, a considerable gap is related to the persistence of antibodies for ASF in surviving animals or piglets, making the interpretation of seropositive detection not obvious. Considering that the knowledge on the duration of maternal antibodies in piglets of sows that survive ASF is not clear and that the only published challenge experiment reports maternal antibodies in piglets of 7 weeks of age and 4 months post-immunization with attenuated ASFV strains [15,16,17,18,19,20,48,49], their central role in the ASF exit strategy should be contextualized. The duration of maternal antibodies of other viral infections of pigs and wild boar provides an indication of the possible time range for maternal antibodies against viruses that may persist in piglets, indicating that the protection from clinical disease may last at least several months in animals recovering from the disease. The classical swine fever virus (an RNA virus) and porcine parvovirus (a DNA virus) maternal antibodies have been shown to last in piglets for 2–4 months [50,51], whereas Aujeszky’s disease virus (a DNA virus) maternal antibodies lasted up to 6 months [21,52], depending on the animal weight. Thus, it may happen to find maternal antibodies for a longer period than expected, increasing the variability associated to seropositivity finding [21].

Concerning the more-appropriate approach for the Sardinian exit strategy, considering the inhomogeneous sampling during the hunting season, the approach based on wild boar density should be the choice. Otherwise, the key role of the accuracy of the data about the abundance of host species in determining the results of the first approach is demonstrated by the example of the NB-HMU: the highest wild boar density (1.16 wild boar hunted/km^2^ vs. 0.75 and 0.77 for the others) generate the highest hunting effort and capturability in this area and, consequently, the possibility to reach more quickly the required number of carcasses. This highlights the fundamental importance of correctly evaluate the local density. On the other hand, the data on hunting bag are very sensitive to the effort and the management of the hunting season, while the use of data on environmental suitability and numerical indices are difficult to pass, especially in the medium–long period [44,45].

Alternatively, several countries characterized by high hunter collaboration and robust hunting bag data should apply the second approach using the amount of wild boar hunted as the estimation of 90% of the dead population [9,11]. If this approach is chosen, a reconstruction analysis is strongly suggested to verify the goodness of the hunting sample with its composition by sex and age classes, including demographic features (i.e., fertility). The hunting sample should, therefore, be compared with demographic matrices, assuming that the compositional quality of the sample is equivalent or comparable to the quantitative one. If the sample correctly reflects the structure by age and sex classes estimated for the population, it must represent an evaluable fraction if the cohort is followed consistently over time.

Finally, a well-planned analysis of cost effectiveness should be carried out, including direct and indirect costs, when a country would implement passive surveillance aimed at detecting an established number of carcasses. The planned activities must be in line with the sustainable costs and main objects of the stakeholders. Furthermore, the relevant role of young and subadult seropositive animals, the detection of which nullifies all of the actions carried out in application of the exit strategy, is a heavy provision to take into account.

## 5. Conclusions

The implementation of an efficacy ASF passive surveillance is a very articulated process, which needs a country-specific evaluation. Thus, a univocal approach is not applicable. Even if the overall number of carcasses to be found within a time frame was established [9], this number is not adequate for all the infected countries. This work has not only scientific value but also practical value given that the online tool *WBC-Counter* developed is the first tool of grate usefulness to lead the choice of the most appropriate strategy country-by-country. Furthermore, it could be used by Veterinary services and policy maker of different countries worldwide at final stage of ASF eradication to make ASF surveillance focused on proving the freedom from ASF.

First, the practical actions to carry out when the passive surveillance have to be put in place should be planned based on country-specific contexts. The number of human patrols, with or without dogs, to be involved during the time frame in order to explore the identified area must first be defined based on the epidemiological situation (i.e., wild boar density, the number of carcasses expected, and the prevalence of ASF).

Second, this organization must be suitable for the proper environmental and management conditions of this area. Considering that both active and passive surveillance must move forward concurrently, and that the success of surveillances depends on each other, not considering the hunting season management could generate an insurmountable bias.

Finally, the target approach must take into account the social context and the resources needed/available in terms of the associated costs. To be successful, passive surveillance requests the employment of several people for a short period, as demonstrated by Desvaux et al. (2022) [13]. The amount of expenditure required to find carcasses in a short time is not minimal nor sustainable for each country currently affected by the ASFV. Alternative approaches must be taken into account.

## Figures and Tables

**Figure 1 viruses-14-01424-f001:**
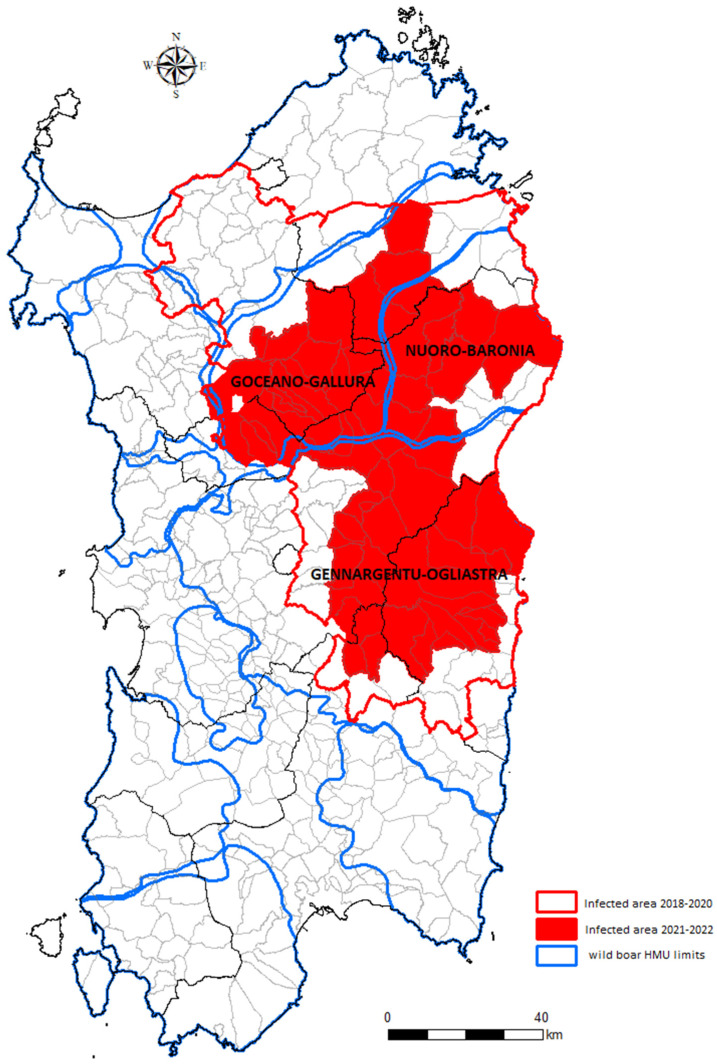
Map showing the limits of the wild boar hunting management units (HMUs), including the Sardinian wild boar infected area. Red outline indicates the old infected area (2018–2020), while the full red area indicated the new 2021–2022 infected area.

**Figure 2 viruses-14-01424-f002:**
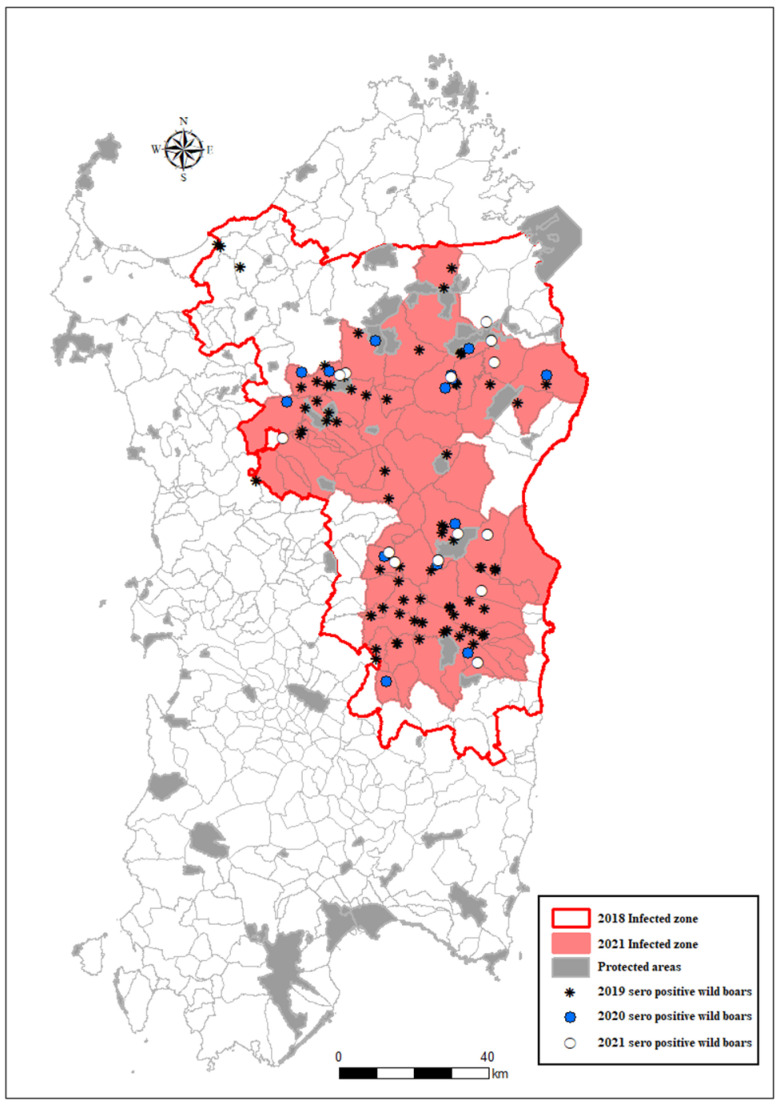
Wild boar-infected zone and the location of seropositive wild boars in Sardinia in 2019–2022.

**Figure 3 viruses-14-01424-f003:**
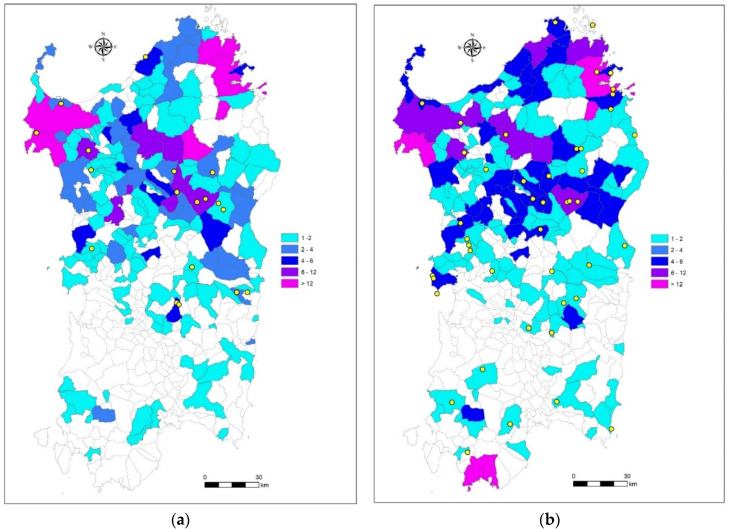
Geographical distributions of the samples taken and tested distinguished by found dead animals (carcasses) and road kills in 2020 (**a**) and 2021 (**b**). Gradient of color indicates the overall number of wild boar killed by road traffic in each common. Yellow dots indicate the places of each carcass findings.

**Figure 4 viruses-14-01424-f004:**
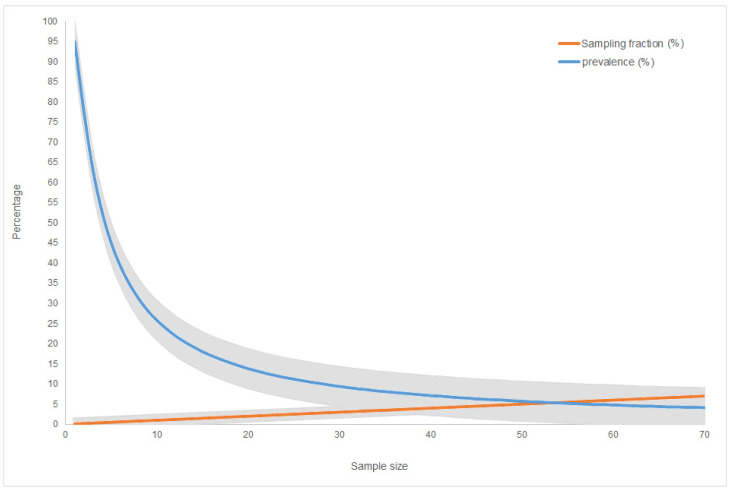
Graphical representation of the minimum ASF-negative carcasses (sample size) corresponding to the sampling fraction (%) of the overall population of the estimated carcasses and the disease prevalence (%) that could be excluded by the carcasses when assuming a maximum overall population of 2000 carcasses. The grey bands represent the 95% confidence limits.

**Table 1 viruses-14-01424-t001:** The eight main EFSA scientific opinion founding [9] are identified by the main scientific key points and summarized as the indicator and the specification of these founding.

	Key Points	Indicator	Specification
1	Land subdivision	Extension of the area (km^2^)	The exit strategy should be evaluated based on a portion of territory (i.e., LAU1, or HMU)
2	Active surveillance performance	Number of months covered by active surveillance, probability of false decision	The exit strategy to be applied in a specific area should be evaluated based on the combination of both active and passive surveillance, during which all hunted wild boar must be ASF tested. The inclusion or omission of active surveillance determines a shorten or longer period to make decisions and strategy performance in terms of the probability of a false decision
3	Number of samples expected—screening phase	Found dead animals expected to be found in a specific area by the extension of the area during the screening phase	During the screening phase, a total of one carcass/1000 km^2^/year should be found. Only found dead wild boar must be counted as valid carcasses, not wild boar killed by traffic accidents
4	Screening phase—month count	Number of months after the last ASFV PCR+ detection in each area	The screening phase starts from the last ASFV detection
5	Screening phase—carcasses found	Number of carcasses found during the screening phase in each area	During the screening phase, passive surveillance aiming to detect at least one carcass/1000 km^2^/year must be applied
6	Number of samples expected—screening phase	Animals found dead expected to be found in a specific area by the extension of the area during the confirmatory phase	During the confirmatory phase, a total of one, two, or six carcasses/1000 km^2^/year should be found. Only wild board found dead must be counted as valid carcasses, not wild boar killed by traffic accidents
7	Confirmatory phase—month count	Duration of the confirmatory phase (months) in each area	The confirmatory phase starts after the screening phase, and its duration depends on the level of confidence and the duration of the screening phase
8	Confirmatory phase—carcasses found	Number of carcasses found during the confirmatory phase in each area	During the confirmatory phase, enforced passive surveillance aiming to detect at least one, two, or six carcasses/1000 km^2^/year must be applied

**Table 2 viruses-14-01424-t002:** ASF seroprevalence in wild boar in the infected area, detected by active surveillance divided by animal age categories and wild boar hunting management units (HMUs), since the 9 April 2019.

Hunting Management Unit *	Age Category	2019	2020	2021	2022
Goceano-Gallura (GG)	adult	17 (3.95)	6 (1.62)	0 (0)	1 (0.12)
	subadult	1 (0.23)	0 (0)	1 (0.23)	1 (0.57)
	young	0 (0)	0 (0)	0 (0)	0 (0)
Nuoro-Baronia (NB)	adult	4 (0.44)	4 (0.75)	6 (0.73)	1 (0.13)
	subadult	1 (0.36)	0 (0)	1 (0.26)	1 (0.41)
	young	0 (0)	0 (0)	0 (0)	0 (0)
Gennargentu-Ogliastra (GO)	adult	27 (1.18)	16 (1.19)	9 (0.66)	0 (0)
	subadult	3 (0.45)	1 (0.11)	0 (0)	0 (0)
	young	0 (0)	1 (1.13)	1 (1.12)	1 (1.5)

* Data are reported as the number of seropositive wild boar detected and the seroprevalence as a percentage by animal age category in each of the three HMUs. Young (0–6 months), subadult (6–18 moths) and adult (>18 months).

**Table 3 viruses-14-01424-t003:** The number of wild boar sampled and tested by passive surveillance in the infected zone divided by carcasses and wild boar killed by road traffic from the 9 April 2019 to 15 June 2022.

	2019 ^¥^	2020 ^¥^	2021 ^§^	2022 ^¶^	Total
**Carcasses ^#^**	37 (4.0)	32 (3.4)	20 (3.7)	9 (2.1)	98
**Killed by road traffic ***	206 (22.1)	195 (21.0)	173 (32.0)	72 (13.6)	646
**Total**	243 (26.1)	227 (24.4)	193 (35.7)	81 (15.7)	744

^#^ Data are reported as number (density per 1000/km^2^) of carcasses found inside the infected zone. * Data are reported as number (density per 1000/km^2^) of wild boar killed by road traffic. ^¥^ Data are reported based on the limits of the infected zone 2018–2020 (brown line, Figure 1). ^§^ Data are reported based on the limits of the new infected zone 2021–2022 (red line, Figure 1). **^¶^** Data are reported based on the limits of the new infected zone 2021–2022 (red line, Figure 2), updated to 15th of June 2022.

**Table 4 viruses-14-01424-t004:** Data estimated by the *WBC-Counter* tool when applying the first approach based on the wild boar density.

HMU	Area Surface (km^2^)	Wild Boar Population	Wild Boar That Died by Hunting	Wild Boar That Naturally Died	Carcasses Expected/Year during the Screening Phase	Carcasses Expected/Year during the Confirmatory Phase
GG	1716	6864	3089	343	3	6
NB	1089	4356	1960	218	2	4
GO	2497	9988	4495	499	5	10
Total area	5302	21,208	9544	1060	10	20

**Table 5 viruses-14-01424-t005:** Data estimated by the *WBC-Counter* tool when applying the approach based on the number of wild boar hunted.

HMU	Wild Boar Hunted during the Hunting Season	Wild Boar That Naturally Died	Carcasses Expected/Year during the Screening Phase	Carcasses Expected/Year during the Confirmatory Phase
GG	1300	144	2	4
NB	1266	141	1	2
GO	1943	215	2	4
Total	4509	501	5	10

**Table 6 viruses-14-01424-t006:** Exit strategy indicators for the three HMUs that include the 2021 infected area, based on the 2% probability of a failure scenario and the estimated number of expected carcasses by the *WBC-Counter* tool using the wild boar density approach. Data are reported by the screening and confirmatory phases, including the starting date (last virus or seropositive young wild boar detection), duration (months), number of carcasses expected during one year, proportion of carcasses expected for the duration of the phase, and the number of carcasses collected.

		GG-HMU	NB-HMU	GO-HMU	Total Infected Area
Screening phase	**Starting date**	29/11/2020	06/01/2020	17/01/2021	17/01/2021
	(dd/mm/yyyy)	9	8	7	9
	Carcasses expected/year	3	2	5	10
	Total carcasses expected	2	1	3	7
	Carcasses found	2	2	3	7
Confirmatory phase	Starting date	02/08/2021	09/10/2020	30/08/2021	30/08/2021
	Months	10	10	11	11
	Carcasses expected/year	6	4	10	20
	Total carcasses expected	5	3	8	17
	Carcasses found	6	4	11	21
**Aims**		Exit strategy completed	Exit strategy completed	Exit strategy completed	Exit strategy completed

**Table 7 viruses-14-01424-t007:** Exit strategy indicators for the three HMUs that include the 2021 infected area, based on the 2% probability of a failure scenario and the estimated number of expected carcasses by the *WBC-Counter* tool using the wild boar hunted approach. Data are reported by the screening and confirmatory phases, including the starting date (last virus or seropositive young wild boar detection), duration (months), number of carcasses expected during one year, proportion of carcasses expected for the duration of the phase, and the number of carcasses collected.

		GG-HMU	NB-HMU	GO-HMU	Total Infected Area
Screening phase	**Starting date**	29/11/2020	06/01/2020	17/01/2021	17/01/2021
	(dd/mm/yyyy)	9	8	7	7
	Carcasses expected/year	1	1	2	4
	Total carcasses expected	1	1	1	2
	Carcasses found	2	2	3	7
Confirmatory phase	Starting date	02/08/2021	09/10/2020	30/08/2021	30/08/2021
	Months	12	12	14	12
	Carcasses expected/year	2	2	4	8
	Total carcasses expected	2	2	5	8
	Carcasses found	6	4	11	21
**Aims**		Exit strategy completed	Exit strategy completed	Exit strategy completed	Exit strategy completed

## Data Availability

All the data are reported in the main text.

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
