# Peer review of "Changes in Estimating the Wild Boar Carcasses Sampling Effort: Applying the EFSA ASF Exit Strategy by Means of the WBC-Counter Tool"

_viruses, 2022, doi:10.3390/v14071424_

Round 1

Reviewer 1 Report

The full review is attached in a separate Word doc.

Author Response

Dear reviewer,

As corresponding author and by all the authors I would like to really thank you for the time spent for this revision, for the valid suggestions which improved our work and for the interest you reserved to this work. Several of your consideration were added to the discussion given that the authors considered them very useful. The manuscript has been completely revised by the English editing service of the MDPI. Furthermore, the paper was updated with the 2022 passive surveillance data until the 15 June.

Specific comments on the online tool WBC-Counter:

Lines 111-113 say: A practical free access tool named WBC-Counter (available at: http://r- ubesp.dctv.unipd.it/shiny/WBC-counter) was developed to automatically calculate the number of needed carcasses.

Despite of the description of the online tool provided in the section of Materials and Methods, there are instructions given on the proper use of the tool (not in the manuscript, nor in the website of WBC-Counter. After trying to play with a tool (online) I have several concerns:

  • It seems that tool can be used only for one approach described in the manuscript (considering wild boar density). There is option to enter data on hunted wild boar.
  • it is not clear what data should be entered in fields: Wild boar density (No/1 sq.km or No/10 sq.km) and Area (ha, sq.m or sq.km). These values shall be added to avoid confusion.
  • it is not clear, why there are two Confirmatory phases (1 and 2) if the manuscript and also EFSA document define only one Confirmatory phase. This could mislead the user.
  • it is not clear, what is a meaning of: No found… (it is always equal to the figure entered in data field of Wild boar density). Explanation is needed.
  • It is not clear, what is the meaning of N found confirmatory 1 and 2 and how these results should be interpreted. Does it refer to the number of carcases that shall be the target (carcasses to be found) during the confirmatory phase?
  • Even trying to use the data from the manuscript (tables 4a and 5a) I did not get the same results in the tool.
  • What is the meaning of value - Sensitivity of the Surveillance System (s1): where it refers to? Same with s2.

Considering the concerns mentioned above the online tool named WBC-Counter shall be improved to make it clearer and user friendly.

R: Dear revisor,

Thank you very much for all these suggestions. We included in the work the link of the versioned app, subject to future updates and enhancements such as those in progress, like any dynamic tool designed to be used and not proposed and forgotten. The WBC-Counter has been updated and now the strategy proposed in the website is in line with the paper, following your suggestions.

Specific comments on the text of the manuscript:

Line 130: the EU Commission decision 2014/709/EU was repealed by Commission Implementing Regulation (EU) 2021/605 in 21 April 2021. Please make changes accordingly.

R: the suggested changes have been made

Figure 1: legend is missing. In a title, what is meant by “limits”? As there are two colours, there could be problems for readers using black-and-white version (printout). Consider changing.

R: Figure 1 has been update with new colours and adequate legend.

Lines 144-145: perhaps specific reference is needed to the National Italian Law.

R: specific reference was added

Line 181: it is not clear what is meant by the word “foundings”. Please consider using other word.

R: we substituted “foundings” with “key points”

Lines 181-182: please provide the reference to EFSA scientific opinion.

R: specific reference was added

Line 183: you refer to Table 1 mentioning seven indicators, but they are eight in Tab.1. Please correct.

R: corrected with eight in the table and in the text

Lines 184: Table 1 does not have a title.

R: the title was added: “Table 1. The eight main EFSA scientific opinion founding [9] are identified by the main scientific key points and summarized as the indicator and the specification of these founding.”

Table 1: Indicators No 4 and No 7: I think they should refer to Confirmatory phase instead of Screening phase.

R: thank you, the refuses were corrected

Line 184: something that should not be there.

R: thank you, the refuses were corrected

Line 199: word “subadu850lt” shall be corrected.

R: thank you, the refuses were corrected

Line 199: please specify how wild boar density in a way it should be used in formula. No.animals / km2?

R: thank you for this suggestion, the phrase was substituted as: “The wild boar density, expressed as number of animals/km2, was used to estimate the number of wild boar population living in an area as…”

Line 216: please specify what do you mean by “this number”?

R: “this number” was substituted with “the wbhunt

Lines 261-262: please indicate the exact period of data provided in the Table 2.

R: The title of the Table 2 was modified as: “Table 2. ASF seroprevalence in wild boar in infected area detected by active surveillance divided by animal age categories and wild boar Hunting Management Units (HMU), since the 9th April, 2019 to 31th December, 2021.”

Table 2: the figure “0 (484)” for the year 2020 should not be correct.

R: thank you very much for this, the refuse was corrected

Lines 265-266: This statement is not clearly seen in the Fig.2.

R: the statement has been changed as: “All wild boar cases detected during 2019-2021 were located inside the infected area mainly clustered in two areas (north and south), as graphically represented  in Figure 2. The cases located in the first visual cluster in the north of the infected area were close to nature parks where hunting is not permitted (blue limits). The cases in the second visual cluster (south) were mainly detected in areas where the presence of illegal free-ranging pigs had been frequently detected.“

Line 270: Please add the period you referring to (perhaps 2019-2021?).

R: all the manuscript was update with 2022 data, thus the period 2019-2022 was specified

Figure 2: There could be problems for readers using black-and-white version (printout). Consider changing.

R: figure has been changed, we hope now the black-and-white version is even more understandable

Line 278: Please indicate the exact period of data provided in the Table 3.

R: all the manuscript was update with 2022 data, thus the period 2019-2022 was specified

Line 287: Expression “filled sessions” is not clear. Please consider rewording.

R: the expression “field sessions” was changed in “passive surveillance activities”

Line 287-288: Can spring season be in May and September?

R: the sentence was changed as “The activities of passive surveillance carried out between May and September (mean temperature 16° C)...”

Lines 355-356: There is a confusion because in a Table 4b the number of wild boar carcasses needed for Exit strategy is 5 and 10 (instead 4 and 8 you mention)? Could you please clarify this difference?

R: we are really sorry for this refuse, now the number of carcasses needed indicates in the text and in the four tables has been update accordingly (Tables 4a,b and 5a,b)

Line 358: is the 12 right figure? Or there should be 15 carcasses?

R: we are really sorry for this refuse, now the number of carcasses needed indicates in the text and in the four tables has been update accordingly (Tables 4a,b and 5a,b)

Lines 469-470: The first sentence of Conclusions should be reviewed to make is clearer.

R: the conclusion was reviewed

Conclusions: As the online tool WBC-Counter has been developed during this study, please consider adding a conclusion on this newly developed tool and it`s usefulness.

R: a sentence on the importance of this new tool was added

Lines 517-519: Acknowledgements – seems that the present text is not written by authors… Please correct.

R: Acknowledgments section was removed

Reviewer 2 Report

This is a very practical case illustrating the difficulty of implementing recommendations from very different environmental situations. 

The application of a theoretical method to a real situation allows to highlight some inconsistencies and will be valuable to authorities to improve their recommendations. This is a very useful paper.

I think the manuscript could be greatly improved by an English-speaking reader.

Some comments/suggestions

lines 84-104 should be summarized: many of these elements are mentioned in the discussion

line 163 : you can clearly name the country (Estonia) on which EFSA based its statement

line 167 : you do not need to cite every institutes having contributed to the Sardinian carcass model (the reference is enough)

line 170 : if relevant you should add in the methods how many field sessions that have been organized based on the carcass model

line 184 Table 1 requires a title

line 199 subadult

lines 279 and 280 per 1000 km² (not per 1000/km²)

line 280 add the corresponding symbol (*) in the table

Figure 3 I do not understand the added-value of this figure : you mix found dead (passive) and road kills (active surveillance), you should at least distinguish between both sources of data. The way I interprete this figure is: yellow dots show found dead and colour codes / sector give the overall number of tested animals (road kills and found dead). This should be more clearly explained if the figure is to be maintained

line 320 Your statement does not take the lenght of the screening and confirmatory phases into account. If both phases are longer than 52 weeks, reproduction occures in between and you cannot sum the samples from 2 different years (30 ASF negative carcasses). As shown in table 5a, it looks that both phases exceed 52 weeks. If I am right in my interpretation, figure 4 should be adapted for an overall population of 1060 carcasses x 2 yrs = 2120 (instead of 2000) and the results adapted.

line 342 If you assume that hunting effort and capturability are comparable from one HMU to another, NB-HMU shows the highest density (1.16 wb hunted / km² vs 0.75 and 0.77 for the others). It is then logical to find more dead wild boar and to reach more quickly the required number of carcasses. This highlights the importance of correctly evaluate the local density and should be more discussed in the discussion chapter.

line 374 The discussion chapter could be structured in two parts: a discussion of the results of your own study, a critique of the EFSA strategy with regard to relevance and feasibility. 

The discussion on the Sardinia study could include elements on how to improve the estimation of densities and the organisation of the carcass search (carcass model, effort quantity,...)

Author Response

This is a very practical case illustrating the difficulty of implementing recommendations from very different environmental situations. 

The application of a theoretical method to a real situation allows to highlight some inconsistencies and will be valuable to authorities to improve their recommendations. This is a very useful paper.

I think the manuscript could be greatly improved by an English-speaking reader.

R: dear review,

Thank you very much for all your suggestions and for the time spent for this revision. The manuscript has been completely revised by the English editing service of the MDPI. Furthermore, the paper was updated with the 2022 passive surveillance data until the 15 June.

Some comments/suggestions

lines 84-104 should be summarized: many of these elements are mentioned in the discussion

R: the paragraph was summarized following your suggestion

line 163 : you can clearly name the country (Estonia) on which EFSA based its statement

R: the reference to Estonia was added

line 167 : you do not need to cite every institutes having contributed to the Sardinian carcass model (the reference is enough)

R: dear review, we completely agree with you. The name of the institutes were deleted

line 170 : if relevant you should add in the methods how many field sessions that have been organized based on the carcass model

R: this information was added in M&M session

line 184 Table 1 requires a title

R: the title was added as: “Table 1. The eight main EFSA scientific opinion founding [9] are identified by the main scientific key points and summarized as the indicator and the specification of these founding.”

line 199 subadult

R: corrected

lines 279 and 280 per 1000 km² (not per 1000/km²)

R: corrected

line 280 add the corresponding symbol (*) in the table

R: corrected

Figure 3 I do not understand the added-value of this figure : you mix found dead (passive) and road kills (active surveillance), you should at least distinguish between both sources of data. The way I interprete this figure is: yellow dots show found dead and colour codes / sector give the overall number of tested animals (road kills and found dead). This should be more clearly explained if the figure is to be maintained

R: the added-value of this figure would be to graphically represents the distribution of the samples. To facilitate the interpretation the legend was update. Please, let us know if you prefer to delate the figure.

line 320 Your statement does not take the lenght of the screening and confirmatory phases into account. If both phases are longer than 52 weeks, reproduction occures in between and you cannot sum the samples from 2 different years (30 ASF negative carcasses). As shown in table 5a, it looks that both phases exceed 52 weeks. If I am right in my interpretation, figure 4 should be adapted for an overall population of 1060 carcasses x 2 yrs = 2120 (instead of 2000) and the results adapted.

R: dear review, you completely right, thus we update the calculation and the figure even if changing the overall population the correction in the figure are not significant

line 342 If you assume that hunting effort and capturability are comparable from one HMU to another, NB-HMU shows the highest density (1.16 wb hunted / km² vs 0.75 and 0.77 for the others). It is then logical to find more dead wild boar and to reach more quickly the required number of carcasses. This highlights the importance of correctly evaluate the local density and should be more discussed in the discussion chapter.

R: the discussion was updated following your interesting suggestions

line 374 The discussion chapter could be structured in two parts: a discussion of the results of your own study, a critique of the EFSA strategy with regard to relevance and feasibility. The discussion on the Sardinia study could include elements on how to improve the estimation of densities and the organisation of the carcass search (carcass model, effort quantity,...).

R: discussion was completely revised, not physically dividing in these two paragraphs, but dividing the discussion as you suggested

Reviewer 3 Report

The problem the world faces today is how to control ASF, but the problem it will face in the future is how to eradicate ASF. In this regard, this study in Sardinia is significant because it is an issue that will be needed in other regions in the future.

Major comment

Careless mistakes are noticeable in English writing as a whole. These include misspellings of words, forgetting to delete words, and typos. And these errors make understanding the text even more difficult. It is strongly recommended that the entire text be carefully read again. In addition, the grammar is sometimes incorrect and there are some parts that are difficult to understand, so I recommend that you send the text to an English proofreader once.

Examples of careless miss

L93: may depends on => may depend on

L108: was describe => was described

L181: foundlings => founding?

L205: Km2 => km2

L251: ArcGIS ArcGIS => ArcGIS

Minor comment

Abstracts

Those familiar with ASF in Europe are well aware of the current ASF situation in Sardinia. However, people in the rest of the world may not be familiar with the current situation. Adding one or two sentences referring to the current situation in Sardinia will make it easier for readers to find the significance of this study.

Main text

L124-125: I think it would be better described as farm biosecurity being reviewed or enhanced.

L142-143: In Figure 1, the island is divided into areas by three different colored lines, but there is no indication of the meaning of each color.

L148: If I am correct, this should be listed as ASFV, not ASF. Please check again to make sure all the words are used correctly.

L162-163: Why mention Northern Europe as a comparison? ASF is widely extended to the Eastern and Southern Europe. Is this because the EFSA Exit strategy was developed taking into account the Baltic States?

L184-185: Forgot to delete it?

Table 1.

4 Number of samples expected-Screening phase:

Isn’t it Confirmatory phase? Also, it would be better to insert after 6.

6,8: km2 => km2

L186-191: It is assumed that authors are referring to the graph in the ASF Exit Strategy report of the EFSA SCIENTIFIC OPINION. Please provide the specific relevant section as a reference.

L195-197: The number of carcass doesn’t correspond with that of table 1. “8 Confirmatory phase - car–ass found". Based on the EFSA report is it not 2 or 6 carcass but 1,2 or 6 carcass?

L196: It is not clear to me how to set the duration of the confirmatory phase; L196 says "which duration is half than the screening phase adopted", though it seems to depend on the level of confidence.

L199: subadu850lt => subadult

L213: Is it correct to interpret that this 45% represents the wild boar population hunted in Sardinia? If so, please state so. It is easy to get confused whether talking in general or limited to the island of Sardinia.

Table 2.

What does the sign next to the year mean? If it is not needed, please remove it.

L266-267: It is imagined that here it means that the distribution of carcasses is visually clustered. I suggest that you clearly state that in the text. It is easy to get confused because L272-274 refers to temporal statistical cluster analysis citing another paper.

L279-282: Annotation and table symbols do not match. Please check.

L285-286: Please reconsider the description in Figure 3. What is the sample category in this case? If each color represents the number of samples collected per region, what is the difference between the yellow dots and colored area? Please also clearly describe the legend in the figure.

L287: It says spring season, but why is September included? Is it just a misstatement or is there an intent?

L495-513: The authors' names should only be their initials.

Author Response

The problem the world faces today is how to control ASF, but the problem it will face in the future is how to eradicate ASF. In this regard, this study in Sardinia is significant because it is an issue that will be needed in other regions in the future.

R: dear review,

Thank you very much for all your suggestions and for the time spent for this revision. The manuscript has been completely revised by the English editing service of the MDPI. Furthermore, the paper was updated with the 2022 passive surveillance data until the 15 June.

Major comment

Careless mistakes are noticeable in English writing as a whole. These include misspellings of words, forgetting to delete words, and typos. And these errors make understanding the text even more difficult. It is strongly recommended that the entire text be carefully read again. In addition, the grammar is sometimes incorrect and there are some parts that are difficult to understand, so I recommend that you send the text to an English proofreader once.

Examples of careless miss

L93: may depends on => may depend on

R: The manuscript has been completely revised by the English editing service of the MDPI.

L108: was describe => was described

R: The manuscript has been completely revised by the English editing service of the MDPI.

L181: foundlings => founding?

R: The manuscript has been completely revised by the English editing service of the MDPI.

L205: Km2 => km2

R: corrected

L251: ArcGIS ArcGIS => ArcGIS

R: corrected

Minor comment

Abstracts

Those familiar with ASF in Europe are well aware of the current ASF situation in Sardinia. However, people in the rest of the world may not be familiar with the current situation. Adding one or two sentences referring to the current situation in Sardinia will make it easier for readers to find the significance of this study.

R: thank you for this point of view, some details on Sardinian context were added in the abstract

Main text

L124-125: I think it would be better described as farm biosecurity being reviewed or enhanced.

R: thank you for this point of view, some details on farm biosecurity were added in the session

L142-143: In Figure 1, the island is divided into areas by three different colored lines, but there is no indication of the meaning of each colour.

R: the legend was updated

L148: If I am correct, this should be listed as ASFV, not ASF. Please check again to make sure all the words are used correctly.

R: yes, corrected

L162-163: Why mention Northern Europe as a comparison? ASF is widely extended to the Eastern and Southern Europe. Is this because the EFSA Exit strategy was developed taking into account the Baltic States?

R: yes you right, the sentence was updated following this suggestion

L184-185: Forgot to delete it?

R: yes you right, the sentence was updated following this suggestion

Table 1. 4 Number of samples expected-Screening phase: Isn’t it Confirmatory phase? Also, it would be better to insert after 6.

R: yes you right, the table was updated following this suggestion

6,8: km2 => km2

R: corrected

L186-191: It is assumed that authors are referring to the graph in the ASF Exit Strategy report of the EFSA SCIENTIFIC OPINION. Please provide the specific relevant section as a reference.

R: the reference was added

L195-197: The number of carcass doesn’t correspond with that of table 1. “8 Confirmatory phase - car–ass found". Based on the EFSA report is it not 2 or 6 carcass but 1,2 or 6 carcass?

R: yes you right, the sentence was updated following this suggestion

L196: It is not clear to me how to set the duration of the confirmatory phase; L196 says "which duration is half than the screening phase adopted", though it seems to depend on the level of confidence.

R: the sentence was a refuse and was deleted

L199: subadu850lt => subadult

 R: corrected

L213: Is it correct to interpret that this 45% represents the wild boar population hunted in Sardinia? If so, please state so. It is easy to get confused whether talking in general or limited to the island of Sardinia.

R: dear review, the 45% could be generalized not only for Sardinia but also for several other countries, as reported by the EFSA opinion and some other papers. Some references were added.

Table 2. What does the sign next to the year mean? If it is not needed, please remove it.

R: removed, thank you

L266-267: It is imagined that here it means that the distribution of carcasses is visually clustered. I suggest that you clearly state that in the text. It is easy to get confused because L272-274 refers to temporal statistical cluster analysis citing another paper.

R: yes you right, the sentence was updated following this suggestion

L279-282: Annotation and table symbols do not match. Please check.

 R: corrected

L285-286: Please reconsider the description in Figure 3. What is the sample category in this case? If each color represents the number of samples collected per region, what is the difference between the yellow dots and colored area? Please also clearly describe the legend in the figure.

R: the added-value of this figure would be to graphically represents the distribution of the samples. To facilitate the interpretation the legend was update. Please, let us know if you prefer to delate the figure.

L287: It says spring season, but why is September included? Is it just a misstatement or is there an intent?

R: the sentence was changed as “The activities of passive surveillance carried out between May and September (mean temperature 16° C)...”

L495-513: The authors' names should only be their initials.

 R: corrected

This manuscript is a resubmission of an earlier submission. The following is a list of the peer review reports and author responses from that submission.